# Digital mental health and peer support: Building a Theory of Change informed by stakeholders' perspectives

Meigan Thomson[1]*, Gregor Henderson[2], Tim Rogers[2], Benjamin Locke[2], John Vines[3], Angus MacBeth[4]

**1** MRC/CSO Social & Public Health Sciences Unit, University of Glasgow, Glasgow, United Kingdom, **2** Togetherall, 250 The Esplanade, Toronto, Ontario, Canada, **3** School of Informatics, University of Edinburgh, Edinburgh, United Kingdom, **4** School of Health in Social Science, University of Edinburgh, Edinburgh, United Kingdom

* meigan.thomson@glasgow.ac.uk

**Data Availability Statement:** The data generated and analyzed from this study are available from the UK Data Service ReShare repository: doi.org/10.5255/UKDA-SN-856193.

## Abstract

Digital Mental Health and Peer support has the potential to bridge gaps in support through its scalability and accessibility. Despite the increasing use of these platforms, there remains a lack of understanding of how they operate in real life, from initial engagement to longer-term impact. We aimed to explore the key inputs, processes, user interactions, assumptions, barriers, facilitators, outcomes, and impacts associated with the use of DMH and peer support platforms by developing a Theory of Change with stakeholders. Stakeholders (n = 77) contributed to the formulation of the Theory of Change through a series of online workshops, focus groups, interviews, and open-ended survey feedback. Workshops were structured to capture information related to aspects of the Theory of Change and to allow stakeholders to provide feedback to improve the diagram. A thematic framework approach was used to analyze transcripts to enable comparisons of factors reported by members, commissioners, and platform staff. Stakeholders identified a variety of factors contributing to initial inputs, processes, outcomes, and impact. Engagement emerged as the most significant barrier to the use of platforms. Motivations for use included filling in gaps in available support, connecting with others and upskilling. Different member types determined how users would interact with the platform which could influence the social response of others. Outcomes were largely positive including provision of a safe online space, improvement in wellbeing, and feeling connected to others. Stakeholders noted impact was harder to identify due to the preventative nature of these platforms but suggested this related to the knowledge of available support, reduction in waiting for support and in referrals, and increasing engagement and uptake of the platforms. Stakeholders identified assumptions regarding internet access as a significant barrier. The Theory of Change illustrated three distinct pathways in digital mental health and peer support. Further research is needed to improve engagement and factors influencing engagement, the member experience and how impact is measured.

**Funding:** This project was funded through a ESRC Impact Acceleration Grant which was awarded to the University of Edinburgh. This grant was used to fund different projects - this project being one of them. The grant ref is ES/T50189X/1, with this project having the unique reference as part of this award: EDI-21/22-P0056. This was used to fund MT, AM and GH time on the project. No additional costs were received for the project. The funder had no role in study design, data collection and analysis, decision to publish, or preparation of the manuscript.

**Competing interests:** GH, BL, and TR are employed by Togetherall. Togetherall assisted with recruitment of participants by sending invitations to the study through their platform and directly to those who commission the platform. Analysis and design was conducted and developed by MT and AM. The remaining authors (MT and AM) declare they have no competing interests.

## Author summary

Following the COVID-19 pandemic there has been a growing demand for mental health resources and support. Online platforms offering education and peer support offer a promising way to access support quickly and alleviate demand on services. Research has shown these platforms can effectively provide support but little is understood about how platforms work from beginning (initial use) to end. Our research explored this by interviewing people who work, develop and use these platforms. Our findings suggest the platforms can be used in different ways such as an educational tool or connecting with other people. We found digital skills and access, interactions with peers, personal expectations impacted whether people engaged with the platform. Those who did use the platform experienced a range of benefits including feeling more connected, improved wellbeing and learning new skills. Use of the platform was also thought to alleviate demand on services. Our study provides new insights into how those who use and provide these platforms understand their use and impact on individuals and mental health resources.

## Introduction

### Background

Global estimates indicate the COVID-19 pandemic led to a 25–27% increase in the prevalence of common mental health disorders like anxiety and depression, with elevated risk for women, young adults, and those with pre-existing health conditions [1]. Common problems reported by people experiencing worsening mental health during the pandemic included increased loneliness, anxiety, and decreased motivation and productivity [2]. Mental health challenges are nested in a wider ecosystem of social and health inequalities experienced within society, perpetuating throughout the lifespan [3]. Moreover, living with mental health issues has been associated with an increased likelihood of having less social support, living with a physical health condition, higher rates of deprivation, and higher premature mortality [4]. There is an urgent need to strengthen and improve mental health systems globally, by improving support and access and engaging in mitigation and management strategies [5].

Digital Mental Health (DMH) offers a scalable approach which can mediate mental health issues and improve access to evidence-based psychological treatments and support, addressing barriers experienced through social and health inequalities [6]. DMH platforms can be delivered via websites, mobile applications, and telehealth. For some populations, this offers quicker and more convenient access to mental health support, addressing potential barriers to access such as waiting lists, cost, and geographical distance [7]. Compared to traditional services, DMH platforms have shown promise in reaching underserved populations, cost-effectiveness, and can ensure the safety of participants of platforms through clearly established risk management protocols and providing pathways to higher intensity interventions [8,9].

Peer support is commonly embedded within these platforms. Peer support involves people sharing knowledge, experiences, and advice with one another [10]. Implementation varies between platforms but is either informal (i.e., naturally occurring between untrained peers), formal (i.e., delivered by a peer who has received some training) or a combination of both [11]. Peer support has been shown to improve social connectedness, confidence, and knowledge of mental health and reduce feelings of isolation for both the recipient and provider [10,12,13]. Furthermore, evidence suggests it develops person-centred skills and has a positive effect on mental well-being whilst being on waitlists [12,14].

DMH-delivered peer support has been found to be more effective in improving mental health in people with worsening mental health symptoms compared to DMH without peer support [15]. Specifically, research has shown the provision of DMH coupled with peer support is associated with improvements in mood and anxiety symptoms [16–20]. Improvements in well-being are similar when delivered via peers or clinicians [21].

Users of DMH platforms report a range of benefits including easier access, anonymity, increased opportunities for self-reflection and connection with others, and availability of advice from people with shared experiences, often promoting hope and empowerment [22–25]. Reviews of the factors influencing engagement with DMH platforms identify key factors including user characteristics (e.g., severity of mental health symptoms, integration of the platform into their daily life and practical skills), user experience of DMH content (e.g. information and features of the platform, feelings of social connectedness and whether the platform is supporting them with their problems), and the technology used to deliver these platforms (e.g. format, privacy and confidentiality concerns) [26,27]. From a commissioner or health professionals' point-of-view, issues around risk and management are commonly reported–including how to control content, prevent further deterioration of mental health, managing and supporting those who present in crisis, the need for training of peer supporters and privacy concerns [19,28,29].

Despite the promise of DMH and peer support to bridge gaps in support, further investigation is needed to fully understand where it can be appropriately integrated into mental health services [29]. Enhancing understanding of DMH platforms across the usage cycle from initial engagement to longer-term outcomes can improve platforms. However, there are challenges in understanding how DMH can be best integrated into the wider mental health system, how users interact with the platform and what the short- and long-term outcomes from DMH and peer support are. One method to understand and explore these issues in DMH and peer support is by developing a Theory of Change.

A Theory of Change (ToC) is one method to explore how and why an intervention operates from beginning to end [30]. ToCs are commonly used as a method to develop, plan, understand, and evaluate how interventions work in the real world [31–33]. It provides an understanding of how an intervention influences behaviour and leads to change by examining the inputs, processes, outcomes, and the broader contextual factors involved. This approach can identify specific conditions under which the intervention is effective, illuminating the connections, assumptions, facilitators, and barriers both within and external to the intervention [34]. By understanding these factors, public health interventions can be adapted and improved to be more effective in their implementation and results [32]. As such a ToC enables identification of gaps in support, challenges in implementation, what if influencing the desired behaviour change and what factors may be influencing this [33]. It can therefore be used as a framework for the intervention itself or as a plan on how to effectively deliver an intervention [35,36]. For example, Lund and colleagues used the ToC approach to identify challenges in mental health support in low-and-middle income countries to support the development of a public health intervention. The ToC helped to identify key challenges including limited resources for mental health support, and high staff turnover (i.e. as a result the time to train new staff could inhibit access to support). To create a ToC, it is crucial to engage a variety of stakeholders who can contribute their experiences and insights, making it an iterative process that involves ongoing stakeholder involvement [33]. By comprehensively considering how and why an intervention works from beginning to end and within the broader social and environmental context, ToCs help pinpoint problems, benefits, and areas where more knowledge and research are needed.

## Objectives

The current study aimed to engage with stakeholders of a DMH platform, Togetherall, to create a ToC for DMH and peer support. We had the following objectives:

1. To identify the key inputs and processes involved in a member of the public using a DMH and peer support platform.

2. To explore how people use and interact with the components of these platforms

4. To consider assumptions, barriers, and facilitators to use of platforms

5. To identify outcomes and impact of DMH and peer support platforms

The study collaborated with Togetherall to construct a practical model illustrating how a DMH peer-support platform operates. This process involved gathering insights from stakeholders who either worked for, commissioned, or used the platform. The resulting perspectives were compiled to create a ToC, enhancing our comprehension of how DMH peer support functions in practice and the benefits it offers.

## Materials & methods

### The Togetherall Platform

Togetherall was selected as a representative of an online digital mental health platform combined with with moderated peer support due to its increasing use in the UK [29]. It has been incorporated into a range of education, healthcare and employment settings therefore provided an opportunity to gain insights from a range of stakeholders. Togetherall is a digital platform offering 24/7 clinically moderated mental health care through directed peer support [29]. The platform aids individuals facing mental health challenges by offering a peer support community, overseen by trained clinicians who moderate the platform and provide one-on-one assistance as needed. This includes enforcing house rules, shaping the content of the community to be supportive and therapeutic, and responding to individuals in distress or crisis who require immediate support. Users have the freedom to share their mental health experiences in real-time and comment on others' posts to provide emotional support and advice. Togetherall is entirely moderated by licensed/registered mental health professionals, called "Wall Guides", who are supported by technology. Posts and interactions are monitored to identify at-risk individuals for harm reduction, to shape community content and to ensure there is a positive and supportive community for members [9,29].

Additionally, the platform provides learning resources, mental health self-assessments, groups focused on key topics and conditions, as well as journaling and goal-setting features [29]. The platform underwent significant development over the past 15 years, with a comprehensive redesign occurring four years ago. It undergoes continuous updates and improvements, with each enhancement informed by the experiences of platform members and the clinical team. It is accessible in various jurisdictions, including the US, Canada, the UK, Ireland, New Zealand, and Australia. The platform is commissioned by a variety of organizations, including higher education, healthcare institutions, local governments, employers, and voluntary sector organizations.

### Study design

The study used a qualitative approach combining stakeholder events, focus groups, individual interviews, and written member feedback to develop and revise the ToC. Procedures and reporting of the qualitative data were conducted in accordance with the Consolidated Criteria

for Reporting Qualitative Research guidance [37]. This study protocol was reviewed and approved by the University of Edinburgh School of Health in Social Science ethics committee, approval number CLPS235. All participants provided consent via an online link which provided a participant information sheet and privacy notice. Consent was gathered for depositing data into an archive for research and teaching.

## Recruitment

Participants for this study were recruited via Togetherall. Togetherall emailed relevant stakeholders to invite them to the study and interested parties signed up to participate using a link included in the email. Recruitment occurred between June and November 2022.

Participants were stakeholders involved in either the development, utilization or provision of the Togetherall platform. These stakeholders included:

1. Togetherall employees, including staff engaged in marketing, design, and the provision of clinical support.

2. Commissioners responsible for making Togetherall accessible to their staff, students, or patients.

3. Members who were actively using the platform for mental health support.

Furthermore, it was a prerequisite for participants to meet specific eligibility criteria to participate. These included being over the age of 18, residing in the United Kingdom, and having the capacity to provide independent consent for their participation in the study.

## Participants

A total of 77 stakeholders contributed to the development of the ToC. This encompassed 17 commissioners and members attending the stakeholder workshops, 7 taking part in the focus group and interview, and 53 anonymous written responses from members in the Togetherall survey. Commissioners included participants from healthcare, education, and employment settings. An overview of participant demographics can be found in Table 1.

## Data collection

As noted above we used a range of resources and methods to develop the ToC.

Two stakeholder online workshops were organized. The initial workshop served as an introduction to the subject and the concept of a ToC. During this workshop, participants worked in breakout sessions to discuss various aspects of the ToC, such as inputs and assumptions.

After the first stakeholder workshop, our research team identified the project needed further input from the Togetherall clinical team, education services, employers and the health sectors, and members of the platform. These groups were subsequently invited to either participate in online interviews or join focus group discussions. Despite attempting to recruit members through the Togetherall platform and the University of Edinburgh, no members signed up to participate in the workshops or interviews. To gain more insight into member experience, we reviewed member feedback collected through a Togetherall survey to inform the ToC. The survey asked members what they liked, disliked or would change about the platform.

Towards the conclusion of the project, we convened a final stakeholder workshop to present the preliminary version of the ToC. Invitations were extended to all individuals who took part in any phase of the project. This workshop provided stakeholders with the opportunity to

**Table 1. Characteristics of participants in the study.**

| | Stakeholder Workshops (n = 17) | Focus Groups & Interview (n = 7) | Togetherall Survey (n = 53) |
|---|---|---|---|
| **Role in digital mental health (n)*** | | | |
| Local Authority | 6 | | |
| Charity | 1 | | |
| Education | 5 | 6 | |
| Togetherall Member | 1 | 1 | 53 |
| Togetherall Staff Member | 5 | 1 | |
| Employer providing the service to staff | | 1 | |
| Gender (female, n) | 12 | 5 | Unknown (anonymous data) |
| Age Group (years, n) | | | |
| 18–30 | 4 | 2 | |
| 31–40 | 6 | 1 | |
| 41–50 | 1 | 1 | |
| 51–60 | 3 | 2 | |
| 61+ | 2 | 1 | |
| Unknown/Not answered | 1 | 0 | 53 |

*some participants had multiple roles

provide feedback on the findings and address any existing gaps in the ToC. A summary of study processes is summarized in Table 2.

## Data analysis

Transcripts from the workshops and interviews/focus groups and data from the Togetherall survey were analyzed using NVivo 12 software [38]. All transcripts were coded by MT and results were discussed and checked by GH and AM. A pragmatic approach was employed to consider the practical uses, limitations, and benefits of using a platform like Togetherall. Transcripts were analyzed using a thematic framework approach [39,40]. This approach enables

**Table 2. Steps in Developing the Theory of Change.**

| | | |
|---|---|---|
| Step 1 | Stakeholder Launch Workshop | Online workshop on Microsoft Teams with a mixture of presentations and breakout rooms to begin developing the ToC |
| Step 2 | The first draft of the ToC Identifying gaps in the ToC and stakeholders we needed to engage with | |
| Step 3 | Interviews and focus groups with stakeholders | Interviews and focus groups were conducted with Togetherall staff, healthcare, education, and employer commissioners |
| Step 4 | Analysis of member written feedback through a Togetherall online survey | |
| Step 5 | Finalizing the first draft of ToC | |
| Step 6 | Stakeholder Close Workshop | Online workshop on Microsoft Teams with a mixture of presentations and breakout rooms to discuss the ToC and identify any gaps/revisions required |
| Step 7 | Finalization of ToC following stakeholder feedback | |

All events and interviews took place on Microsoft Teams and were auto-transcribed and recorded. Workshops lasted 2 hours and interview/focus group lasted approximately 60 minutes.

comparisons to be made between groups (i.e., Togetherall staff, commissioners, members). The framework approach required the researcher(s) to familiarize themselves with the data by reading transcripts and, thereafter, drafting a preliminary coding framework. The preliminary coding framework was reviewed against the first stakeholder event and focus group which led to further refinement and finalizing of the coding framework. A combination of deductive (i.e., structured according to the conventional ToC format) and inductive coding (i.e., data-driven) was used. Subsequently, the coding framework was applied to all data (i.e., transcripts and written member feedback). For the written member feedback, answers to questions asking what members liked, disliked, or would change about the platform were included in the analysis. To facilitate comparison among the groups, the data were organized into a framework matrix in NVivo 12, enabling the creation of summaries for each code to identify the inputs, processes, and outputs reported by each group [41].

### Reflexivity

Stakeholder events were led by MT, GH, and AM, who have experience in research, practice and policy-making for mental health, the use of digital platforms, and the role of psychological interventions within this. Given the research teams' experience in the area, careful consideration was given to minimize how their positionality (i.e. interest and work in the DMH sphere) would impact participant groups [42]. Each event had a well-defined guide for facilitators to follow in addressing the components of the ToC. Furthermore, to minimize bias, Togetherall employees were placed in breakout rooms with a facilitator not affiliated with Togetherall.

Interviews and focus groups were conducted by MT, a mixed methods researcher with experience in applied psychology research and qualitative methods. To ensure objectivity, the interview schedule was reviewed and informed by input from the research team (AM, GH) and the existing literature. Additionally, feedback from stakeholders at the launch event was used to identify knowledge gaps, which in turn informed the development of subsequent interview and focus group schedules. Throughout the coding and interpretation phases, MT engaged in discussions with the research team. During the final stakeholder event, the complete ToC was presented, and any gaps or uncertainties identified by the research team from the data were highlighted and discussed. These methods maximized stakeholder input to the ToC, ensuring a clear interpretation of the results.

## Results

### Key findings

Fig 1 shows the full ToC. The ToC identified 3 distinct trajectories for members, commissioners, and the platform. The ToC shows the different inputs, processes, and outcomes for each trajectory, culminating in the impact of these platforms. Actions undertaken in each pathway often contributed to the outcomes of other pathways (e.g., Members teaching others how to use the platform leads to increased engagement which supports funding applications to have the platform available in the future).

### Inputs

The ToC identified a multitude of inputs into the use of the platform. Inputs refer to the processes and activities which occurred before a member signed up or their first-time use of the platform.

This began with the appropriate content, design and maintenance of the platform, training of the staff, and engagement with commissioners to promote the adoption of the platform.

## Digital Mental Health & Peer Support -Theory of Change

**Fig 1. TOC: Theory of Change of Digital Mental Health and Peer Support Platforms.**

Togetherall staff identified the importance of using clinically driven content (i.e., evidence-based resources and information) in supporting members using the platform to learn and understand their mental health:

> "Our courses and resources they're all clinically driven. . . making sure that information is correct and accurate, because there's a lot of information available online regarding mental health which isn't, which can obviously have a massive impact on how people might perceive they are feeling and therefore yeah impact their mental health negatively" (Togetherall staff member, Stakeholder Launch event)

Characteristics of the platform such as anonymity, having access to peer support and mental health resources, as well as the physical appearance of the platform were highlighted as key motivators for commissioners and members to engage with the platform. As well as the appearance of the platform motivating use, commissioners reported offering the platform to students/clients/employees/patients to bridge perceived gaps in support, which became a more evident need during the COVID-19 pandemic:

> "I think from a for kind of mental health support during the pandemic. It became much more necessary and the wrap-around care has become a real expectation from our student body" (Commissioner in education, Focus Group)

Commissioners also felt motivated to use the platform to upskill themselves due to perceived gaps in training and continued professional development. They hoped using the resources they refer their students/clients could improve their understanding and competency in supporting them with their mental health:

*"Even for the staff to another thing that's helpful is that is the modules and the training. . . to be able to learn more and be able to feel like they're upskilling themselves to support learners."* (Commissioner in Education, Focus Group)

Participants identified a few barriers to initiating the use of the platform. These encompassed challenges with promoting uptake and continued engagement with the platform. Commissioners in the education sector highlighted challenges in promoting engagement with young people and males with the platform:

*"We have a lot of problems trying to engage males in the best of times cause they just won't"* (Commissioner in Education, Focus Group)

While commissioners found it challenging to pinpoint reasons for difficulties in uptake, some linked it to issues with IT literacy, accessibility (i.e., the absence of a mobile application or reminders to use the resource and digital poverty), lack of evidence to support funding of the resource and competing resources:

*"I think this idea of Internet poverty is one that we're coming across that we were not coming across when we started doing this. . . I suppose the more that they used the service the more that they be online, the more that be online, the opportunity cost is then they cannae [can not] do other things online because they've used their data source."* (Commissioner in Education, Stakeholder Close Event)

*". . .for me to access funding, I had to prove that this was something that young people in Orkney wanted and would use."* (Commissioner in Health and Education (educational psychology), Stakeholder Launch event)

*"We've actually decommissioned the platform this year because it wasn't working. I don't know whether or not that was because there was too many things on offer."* (Commissioner in Education, Stakeholder Launch Event)

## Processes

Processes referred to activities which participants reported doing *during the use* of the platform. How members used and their thoughts on the platform were monitored by Togetherall through analytics and survey feedback. Members could use the platform in a multitude of ways including interacting with other members through direct messages and public posting, using the learning resources and self-assessments to explore and understand their mental health, and journalling.

Members indicated that initial impressions and interactions with the platform influenced their future engagement. If users found it challenging to use, engage with other users or had a negative experience this led to dropout or worsening symptoms:

*"I suppose if people are sharing things that are difficult, that are sort of beyond our beyond anyone's control in the general population, they could kind of get each other down in a spiral. . .And this happened to me and somebody else just adding and adding and adding. And there's nobody to actually lighten that or give them tools to manage it."* (Stakeholder in Education and Member, Focus Group)

*"I wasn't frustrated by anything in particular, it was a little saddening that the wall bricks were all full of people's struggles and pain and seeing that would be quite overwhelming for me" (Member, Togetherall Survey)*

To overcome this barrier, some commissioners reported seeking out training opportunities for students to learn how to use and navigate the platform. This included having peers training one another or Togetherall staff:

*"So we did ask [Togetherall staff member] to come and do an assembly with young people to show them. . . to show them how to navigate the site more easily, and we're hoping that will lead to young people using it a little bit, a little bit more effectively." (Commissioner in education, focus group)*

Participants discussed in-depth the role of the peer element in the platform. They noted a cycle between posting, feedback from the community and whether this fulfilled expectations and elicited positive feelings. They noted how "social contagion", where viewing other's posts prompted members to engage more and post to the platform. Stakeholders perceived this as a positive way to build community and interactions on the platform:

*"I think, as more people share, it's almost encourages other people. It's like people follow behaviours and patterns, and they're interesting things is on the platform. . . Sometimes people can create bricks. Sometimes people can share posts and we observe that in some cases, not all the time. But if one person creates a brick, chances are the next person that shares would create a brick. Because they're following behaviours they've seen. . . And I think it's because people are observing what's going on the platform, seeing how people are responding and then they're getting involved" (Togetherall staff member, Stakeholder Launch Event)*

The ToC identified that commissioners often engaged with the platform in the same way as members (e.g., using learning resources and interacting with other members) to either support their learning/training in mental health or to be members themselves. Across members and commissioners who used the platform, 3 distinct user types were identified. This included users who actively engage with the community (i.e., post, read and respond to others posts), those who passively engage with the community (i.e., read member posts but do not post themselves), and those who are absent from the peer support element of the platform (i.e., only engage with non-social aspects of the platform)

*"In some cases, people just come what they wanna do is like journal set a goal read some content do some courses and then some people come on they don't necessarily share within the community but they want to one conversations going on in the background with people that they've connected with and then there are some people who come and put it out there to the community" (Togetherall staff member, Stakeholder Launch Event)*

Commissioners and Togetherall staff highlighted the importance of the management of the platform while members used it as key to maintaining a safe online space. This included having clear mechanisms and protocols in place to monitor member posts to ensure appropriate escalation and safeguarding procedures were in place where necessary to support those experiencing crisis:

*"What really sold it for us was the the 24/7 change protection for young person in safeguarding arrangements and that we we were contacted to say that a young person had an active suicide plan and you know. . . they were able to to to track back and [Police Service] were informed and and we entered the Internet and there was an active seriously plan that young person. So from my point of view that was a probably life saved you know so then we start thinking do I continue with this service but what happens if that service is not there and that's a real challenge." (Commissioner in Local Authority, stakeholder close event)*

Furthermore, the platform considered how to maintain a safe place for those not experiencing crisis by managing exposure to triggering or upsetting material from other member posts:

*"We get people, saying how dare you intervene and not allow me to post everything absolutely anything that I want. I would say to those individuals and to anybody asking is the reason that we don't allow people to talk about [lists triggering material]. . . isn't allowed. Those house rules we. . . adhere to very strictly. . . that may be your right elsewhere in society, but it's not your right on this platform because we have a duty to other's safety as well." (Togetherall staff member, Interview)*

## Outcomes

Outcomes from the activities and processes included both short and long-term. Platform processes led to the provision of a safe online community and crisis management of members. This alongside understanding member experiences through monitoring and keeping resources available and up to date led to longer-term improvements in the platform. Commissioners noted the feedback shared by Togetherall could be used to inform their priorities:

*"On the analytics side of things like seeing what courses people are enrolled in really helps you see what the concerns of your student population are. . . lots of people have done the self-assessment for anxiety, for example, maybe we want to up our messaging in this area" (Commissioner in Education, Focus Group)*

Such evidence could also justify to funders the need for such a service to be funded for their cohorts:

*"I use that to you know example too, I can argue for continued function funding that that could have saved one person's life" (Commissioner in Education, Stakeholder Launch Event)*

Most long and short-term outcomes identified from the data related to improvements in the well-being of the member. These included building self-management skills and support networks through providing and receiving support, improved mental health literacy, and being exposed to different life experiences. Stakeholders believed this ultimately led to an improvement in overall symptoms. They stated the key to these was the sense of belonging such platforms bring:

*". . .saying to people there is peer support on Togetherall like you can find your tribe, there's thousands of people. It's not just the institution and that way you know you can actually maybe get understanding and recognition and maybe normalization of those difficult experiences" (Commissioner in Education, Focus Group)*

## Impact

Stakeholders identified 5 potential key impacts of the use of the platform. These were:

1. Increasing uptake and engagement

2. Filling in gaps in available support

3. Preventing worsening mental health

4. Reducing referral to services (due to earlier and more accessible support)

5. Reassuring members and commissioners that there is a resource available 24/7

Participants found it challenging to identify these impacts of DMH and peer support, and they are speculative. They noted this was due to the challenge of measuring future events which do not occur (i.e. prevention of worsening symptoms of suicide):

> *"I think it's that point around how can you almost measure what you prevent?" (Togetherall staff member, Stakeholder Launch Event)*

## Assumptions & Contextual Factors

Several assumptions, barriers and contextual factors were highlighted. Stakeholders noted DMH tools assume a level of technological literacy of users of the platform. Platforms and commissioners also make assumptions related to age, accessibility (i.e., everyone has internet access and internet-ready devices), and members understand data handling and safeguarding procedures. With the changing economic climate, it was noted many members are having to prioritise food or heat over internet access:

> *"I speak to people who told me that they have soup kitchens, breakfast clubs. . .people who are having to make decisions about Internet access. I mean, it's something that we do take for granted" (Commissioner in Education, Stakeholder Close Event)*

When discussing member assumptions, stakeholders largely attributed this to their expectations around social interactions with peers. This included expecting anonymity, to engage with people with similar experiences and for peers to be more engaged than in-person peers:

> *"But something that's not yet been mentioned is the nonjudgmental element. . . young people were saying they felt their own peer groups they were afraid of them being fed up of them talking about their problems. So they stopped talking to their immediate peers and but they would, they would feel in a peer group where people shared similar issues that they wouldn't be judged in the same way." (Commissioner in Health & Education, Stakeholder Launch Event)*

## Discussion

### Principal results

The ToC identified three distinct pathways that influence outcomes and the impact of DMH and peer support. It highlighted the role of the platform itself in building clinically informed and visually appealing content to promote engagement with the platform. Engagement was promoted through signposting and provision of the platform by commissioners, understanding and ability to use the platform, and experiences while using the platform. Discussions

indicated that negative experiences (i.e., finding the platform overwhelming or experiencing negative social interactions) led to disengagement. Recent review work similarly found experience of the platform (i.e., appearance and technology features) influenced engagement [43]. Mirroring our findings, the same review found a range of person-level (e.g., age, personality, and symptom severity), intervention-level (e.g., incentives, gamification, and human support), and system-level factors (e.g., organizational, and social barriers) as key factors as to whether an individual will use a platform.

Activities and processes identified how users of the platform engaged with resources and how the platform monitored the platform to maintain a safe online environment. Togetherall staff and commissioners highlighted that clear protocols were in place to safeguard users, forming an integral part of the platform and used as evidence of the integrity of the platform to support future investment. These processes could be appropriately adapted to suit the user environment (e.g., employer, university, healthcare setting). With the growing use of digital resources in healthcare, there is a need for platforms to employ such procedures for online safety. It has been noted that DMH providers have ethical responsibilities to ensure accountability (i.e., of the support/care provided through the platform), transparency (i.e., of the procedures in place to both users and commissioners), and inclusiveness (i.e., platforms are available to everyone irrespective of demographic characteristics) [44].

Stakeholders suggested there are different member profiles which determine how they interact with the platform: those who use education materials only, those who observe social interactions and those who actively interact with others on the platform. Furthermore, the ToC found commissioners may use the educational courses as a form of professional training. These results mirror existing work on motives for using social platforms for health conditions, suggesting patients use social media platforms to engage with social support, exchange advice and increase their knowledge [45]. Our findings also similarly suggested members use the platform to seek shared experiences and understanding but in circumstances where responses from peers were unexpected or absent this could lead to disengagement.

Specifically related to peer support, our participants reported that over time interactions became reciprocal–i.e., when others posted it prompted more posts from other members. Stakeholders identified peer support as an important element of the platform, opening members up to different types of experiences, allowing the development of person-centred skills through providing and receiving advice, building a support network, and improving symptoms. A review of peer support similarly found associations with improved mental health, including greater happiness, self-esteem, development of effective coping strategies and reductions in depression, loneliness, and anxiety [46]. That said, our stakeholders did note possible negative implications of members feeling unable to provide appropriate support to others, learning or being exposed to unhealthy coping habits, or not receiving the responses from peers that they expected.

Furthermore our study highlighted wider contextual factors as inhibiting engagement. This was in the form of digital literacy, inclusion and poverty. Stakeholders reasoned a degree of digital literacy was key as an initial input to the platform. Members who had a higher level of understanding of how to use the platform would continue to engage but those who experienced difficulty in accessing or navigating the website would disengage. Pertinantly, stakeholders reported internet provision (i.e. access to wifi or data or having an internet compatible device) was a barrier to uptake of the platform. Particularly in members who could not afford their own data and devices due to competing monetary commitments (e.g. energy bills).

Stakeholders found it challenging to identify impacts due to the preventative nature of these platforms. They suggested impacts related to knowledge of support being available, reduction in waiting list referrals, and increased uptake and engagement with these resources. Although

difficult to determine the true impact of these platforms, research does show the presence of these interventions does improve anxiety and depression symptoms than no intervention, particularly with the presence of higher-level supervision and monitoring to safeguard users [47].

## Strengths & limitations

This study provided a coherent ToC overview of the inputs, processes, outcomes, assumptions, and contextual factors associated with the use of DMH and peer support platforms. By focusing on Togetherall, the study is enriched by context and specificity. Nevertheless, we do acknowledge this may influence the generalizability of the ToC components to other DMH platforms. Nonetheless, we have found the results to be consistent with existing evidence around DMH platforms. We highlight areas where DMH platforms can improve (e.g., engagement and uptake) and gaps in our knowledge (e.g., longer-term impact). Importantly, the ToC was informed by many key stakeholders in DMH and peer support (platform staff, commissioners in health, education and employment, and members). The ToC can be used to understand areas for improvement for digital platforms themselves (i.e. focusing on engagement and integration) and to identify gaps for further research (i.e. understanding more clearly member experiences during use). The stakeholders in this research do suggest these platforms can be successfully incorporated into mental health support systems and the benefits this can offer for services (e.g. alleviating demand) and members (e.g. improving mental health awareness and understanding). Notably, the research does reveal key contextual influences such as digital poverty, literacy and inclusion which could be used to inform decisions at a policy level. However, we acknowledge the challenges faced in recruiting members of the DMH platform to the qualitative aspects of this study. This may relate to the anonymous nature of these platforms. To ensure members still had input and a voice in our ToC, we analysed feedback collected by Togetherall on the likes and dislikes of the platform. Our relative lack of member interviews, as compared to commissioners and staff, means that more work is needed to fully understand how users interact with the platforms and identify what leads to further engagement or disengagement. While trying to understand processes and activities, our stakeholders focused largely on community-level interactions, leaving insights into the role of non-community-level interactions with the platform (e.g. direct messaging) absent from the ToC. This may be due to our difficulty in recruiting members to gather these insights or commissioners/platform staff perceive the community-level interactions as more impactful. Finally, our stakeholders found it challenging to understand and explain what the impact of these platforms was. They described subjective impacts around connectedness and well-being but noted more data and a consensus on how to measure outcomes and impact is needed.

## Future research

The ToC highlights gaps in our understanding and implementation of DMH and peer support. The biggest barrier and challenge stakeholders noted was engagement. Considering different ways and barriers to engagement could enhance the uptake of platforms. For example, seeking to better understand from members of platforms how to best balance notifications (timing, type, and purpose) within such platforms. Our stakeholders noted the resources on the platform led to increases in mental health literacy, perhaps incorporating early introductory tools to enable participants to recognize their feelings, and obstacles and how the platform could support this could reduce the pressure on the user to navigate to resources independently. An example of this may be including tools to help users to recognize their emotions and what language can be used to describe this such as Plutchik's "Wheel of Emotions" [48]. Similarly

platforms and researchers need to consider how to make these resources more accessible to those who struggle with digital literacy or are experiencing digital poverty.

Further research is also needed to explore fully the user's experience of the platform. Particularly examine more closely how peer support relationships form, what they mean to the user, and what constitutes good and bad peer support.

A key challenge for our stakeholders was identifying the best way to measure outcomes and impact due to the perceived preventative nature of the platform and members using the platform for different reasons. More research is needed to identify and develop the most appropriate ways to measure outcomes whether it is linked to improvements in mental wellbeing scores, time on the platform, or uptake. Insight into the impact of such platforms being introduced into mainstream services (e.g., primary care or education) through comparison of referral to in-person services before and after introduction could determine whether these platforms should be situated alongside traditional services or as a substitute in some cases.

## Conclusions

The current study developed a theory of change in digital mental health and peer support through engagement with stakeholders. Stakeholders identified a variety of inputs, activities, outcomes, impacts, barriers, and assumptions. Three distinct pathways were identified showing the unique roles of the platform, members, and commissioners at each stage of the ToC. The study highlighted the need for further research into factors influencing and promoting engagement, member's experience of peer support and how impact can be accurately measured.

## Supporting information

**S1 Checklist. COREQ checklist.**
(DOC)

## Acknowledgments

We would like to thank all the stakeholders who participated in this research, and the Togetherall team who supported recruitment for the study.

## Author Contributions

**Conceptualization:** Gregor Henderson, Tim Rogers, Benjamin Locke, John Vines, Angus MacBeth.

**Data curation:** Meigan Thomson.

**Formal analysis:** Meigan Thomson, Angus MacBeth.

**Funding acquisition:** Gregor Henderson, John Vines, Angus MacBeth.

**Investigation:** Meigan Thomson, Gregor Henderson, Angus MacBeth.

**Methodology:** Meigan Thomson, Gregor Henderson, John Vines, Angus MacBeth.

**Project administration:** Meigan Thomson.

**Resources:** Meigan Thomson, Gregor Henderson, Tim Rogers, Benjamin Locke, Angus MacBeth.

**Software:** Meigan Thomson.

**Supervision:** Angus MacBeth.

**Validation:** Angus MacBeth.

**Visualization:** Meigan Thomson.

**Writing – original draft:** Meigan Thomson.

**Writing – review & editing:** Meigan Thomson, Gregor Henderson, Tim Rogers, Benjamin Locke, John Vines, Angus MacBeth.

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
