## [Decision Letter · Decision Letter 0]

13 Mar 2024

PDIG-D-24-00062

Digital mental health and peer support: Building a Theory of Change informed by stakeholders’ perspectives.

PLOS Digital Health

Dear Dr. Thomson,

Thank you for submitting your manuscript to PLOS Digital Health. After careful consideration, we feel that it has merit but does not fully meet PLOS Digital Health's publication criteria as it currently stands. Therefore, we invite you to submit a revised version of the manuscript that addresses the points raised during the review process.

Please submit your revised manuscript within 60 days May 12 2024 11:59PM. If you will need more time than this to complete your revisions, please reply to this message or contact the journal office at digitalhealth@plos.org. Please include the following items when submitting your revised manuscript:

We look forward to receiving your revised manuscript.

Kind regards,

Calvin Or, PhD

Section Editor

PLOS Digital Health

Journal Requirements:

1. Please send a completed 'Competing Interests' statement, including any COIs declared by your co-authors. If you have no competing interests to declare, please state "The authors have declared that no competing interests exist". Otherwise please declare all competing interests beginning with the statement "I have read the journal's policy and the authors of this manuscript have the following competing interests:"

If you did not receive any funding for this study, please simply state: “The authors received no specific funding for this work.

3. Please provide separate figure files in .tif or .eps format only and remove any figures embedded in your manuscript file. Please also ensure that all files are under our size limit of 10MB.

4. We noticed that you used "unpublished" in the manuscript. We do not allow these references, as the PLOS data access policy requires that all data be either published with the manuscript or made available in a publicly accessible database. Please amend the supplementary material to include the referenced data or remove the references.

Additional Editor Comments (if provided):

I see the authors mentioned that COREQ was used. It would be great if the authors could double-check the contents to ensure that the framework was closely followed. Please see the following:

https://www.equator-network.org/reporting-guidelines/coreq/

Please provide information about participant inclusion and exclusion criteria.

Reviewers' comments:

Reviewer's Responses to Questions

**Comments to the Author**

1. Does this manuscript meet PLOS Digital Health’s publication criteria? Is the manuscript technically sound, and do the data support the conclusions? The manuscript must describe methodologically and ethically rigorous research with conclusions that are appropriately drawn based on the data presented.

Reviewer #1: Yes

Reviewer #2: Yes

Reviewer #3: Partly

2. Has the statistical analysis been performed appropriately and rigorously?

Reviewer #1: N/A

Reviewer #2: N/A

Reviewer #3: Yes

3. Have the authors made all data underlying the findings in their manuscript fully available (please refer to the Data Availability Statement at the start of the manuscript PDF file)?

Reviewer #1: Yes

Reviewer #2: Yes

Reviewer #3: Yes

4. Is the manuscript presented in an intelligible fashion and written in standard English?

Reviewer #1: Yes

Reviewer #2: Yes

Reviewer #3: Yes

5. Review Comments to the Author

Reviewer #1: The article is of great interest to readers in showcasing an approach that augment other studies around digital mental health. In a post pandemic era such research is needed to address a significant unmet clinical need. 

The authors suggested the reporting of results followed the COREQ checklist yet identified a ‘pragmatic approach’ to analysis. I would have liked to have seen a reference to the methodological orientation that underpins the study and wondered if this should be grounded theory? A number of references in this section (data analysis) cannot be followed up (dead ends) and make it harder to fully critique the pathway chosen for the analysis.

Reviewer #2: Thank you for this well written submission. I found this easy to follow and rather self-explanatory. I did find table 2 difficult to follow however, and think this could be presented in a more accessible/typical way. Also, I think you could stress the importance and potential impact of this work far more than you do. Although you talk about future research, I don't think you sell the strengths of this as well as you could.

Reviewer #3: This research is interesting and provides a comprehensive qualitative analysis and reporting of results section. The introduction section would benefit from further description of Theory of Change- more literature showing previous use/effectiveness. A clearer rationale for the research is needed, along with clear contributions to literature outlined in the introduction. It is unclear as to why the platform Togetherall was chosen for this study above other DMH platforms. Previous literature using Togetherall would help make this justification. Participant section should be in Methods, survey is mentioned but no other details given. Digital literacy is briefly mentioned in the results but needs to be picked up again in the discussion section along with how digital inclusion has been acknowledged. Thank you for your submission.

6. PLOS authors have the option to publish the peer review history of their article (what does this mean?). If published, this will include your full peer review and any attached files.

**Do you want your identity to be public for this peer review?** For information about this choice, including consent withdrawal, please see our Privacy Policy.

Reviewer #1: No

Reviewer #2: No

Reviewer #3: No

---

## [Decision Letter · Decision Letter 1]

26 Apr 2024

Digital mental health and peer support: Building a Theory of Change informed by stakeholders’ perspectives.

PDIG-D-24-00062R1

Dear Dr. Thomson,

We are pleased to inform you that your manuscript 'Digital mental health and peer support: Building a Theory of Change informed by stakeholders’ perspectives.' has been provisionally accepted for publication in PLOS Digital Health.

Best regards,

Calvin Kalun Or, PhD

Section Editor

PLOS Digital Health

Reviewer Comments (if any, and for reference):

Reviewer's Responses to Questions

**Comments to the Author**

1. If the authors have adequately addressed your comments raised in a previous round of review and you feel that this manuscript is now acceptable for publication, you may indicate that here to bypass the “Comments to the Author” section, enter your conflict of interest statement in the “Confidential to Editor” section, and submit your "Accept" recommendation.

Reviewer #1: All comments have been addressed

Reviewer #2: All comments have been addressed

Reviewer #3: All comments have been addressed

2. Does this manuscript meet PLOS Digital Health’s publication criteria? Is the manuscript technically sound, and do the data support the conclusions? The manuscript must describe methodologically and ethically rigorous research with conclusions that are appropriately drawn based on the data presented.

Reviewer #1: Yes

Reviewer #2: Yes

Reviewer #3: Yes

3. Has the statistical analysis been performed appropriately and rigorously?

Reviewer #1: N/A

Reviewer #2: Yes

Reviewer #3: N/A

4. Have the authors made all data underlying the findings in their manuscript fully available (please refer to the Data Availability Statement at the start of the manuscript PDF file)?

Reviewer #1: Yes

Reviewer #2: Yes

Reviewer #3: Yes

5. Is the manuscript presented in an intelligible fashion and written in standard English?

Reviewer #1: Yes

Reviewer #2: Yes

Reviewer #3: Yes

6. Review Comments to the Author

Reviewer #1: I am satisfied with the additional work undertaken by authors and recommend the manuscript is now accepted for publication

Reviewer #2: Thank you for your revisions. There are still some formatting issues e.g. tables are incorrectly numbered but I expect these will be edited.

Reviewer #3: Thank you for re-submitting your paper, and responding to the comments made. I believe the manuscript is now much improved and ready for publication.

7. PLOS authors have the option to publish the peer review history of their article (what does this mean?). If published, this will include your full peer review and any attached files.

**Do you want your identity to be public for this peer review?** For information about this choice, including consent withdrawal, please see our Privacy Policy.

Reviewer #1: No

Reviewer #2: No

Reviewer #3: No
